# Synthesis of Di-Block Copolymers Poly (Propylene oxide)-block-Poly (9-(2,3-epoxypropyl) Carbazole) via Sequential Addition of Monomers in One Pot

**DOI:** 10.3390/polym13050763

**Published:** 2021-02-28

**Authors:** Lorella Izzo, Paola Lisa, Olga Sacco, Stefania Pragliola

**Affiliations:** 1Dipartimento di Biotecnologie e Scienze della Vita, Università degli Studi dell’Insubria, 21100 Varese, Italy; lorella.izzo@uninsubria.it; 2Dipartimento di Chimica e Biologia, Università di Salerno, 84084 Fisciano, Italy; lisa91paola@gmail.com (P.L.); osacco@unisa.it (O.S.)

**Keywords:** anionic polymerization, di-block copolymers, carbazole, living polymerization

## Abstract

Polypropylene oxide (PPO) and poly(9-(2,3-epoxypropyl) carbazole) (PEPK) di-block copolymers are prepared in one pot via sequential monomer addition by using *i-*PrONa/*i-*Bu_3_Al as an anionic catalytic system. An almost 100% monomer conversion is obtained, and the length of each block is controlled through the monomer/catalyst ratio used. Copolymer molecular weights are quite close to theoretical values calculated assuming the formation of one polymer chain per catalyst; therefore, it is hypothesized that the polymerization reaction proceeds with a living character. The synthesis appears to be particularly efficient and versatile. The calorimetric properties of copolymers obtained in this work are remarkable, since they show two distinct *T*_g_ values, corresponding to the PPO and PEPK blocks. The optical measurements of di-block copolymers show more analogous features than those of PEPK homopolymer. Copolymer solution emission spectra just exhibit isolated carbazole fluorescence, whereas in the solid state, film spectra show excimer fluorescence.

## 1. Introduction

The synthesis of polymers with pendant carbazolyl groups has attracted great academic and industrial interest thanks to the unique spectroscopic properties that they exhibit [1,2,3,4,5,6,7]. Starting from polyvinylcarbazole (PVK) [8], different kinds of polymers with pendant carbazolyl groups, i.e., polyacrylate [9,10], polymethacrylate [11,12,13,14], polystyrene [15,16,17], and polyolefin [18,19,20,21] derivates, have been reported. Poly(9-(2,3-epoxypropyl) carbazole) (PEPK) has been widely studied because of its high carrier and hole drift mobility, too [22,23,24,25,26]. As regards photoconductive properties of PEPK, they are similar or superior than those of PVK [1]. In recent years, block copolymers presenting polymeric blocks containing pendant carbazolyl groups have also gained much attention because of their potential ability to form ordered structures through self-assembly processes, which makes them promising materials for applications in optoelectronics. For example, they could be employed as light-emitting or organic photo-refractive materials. By tuning the chemical nature and length of each block, block copolymers with peculiar chemical, physical and optoelectronic properties can be suitably obtained. To this aim, controlled polymerization techniques, i.e., controlled radical or living anionic methods, have to be employed. A variety of research papers describe the synthesis of carbazole-containing block copolymers using controlled radical polymerization [27,28,29]. By living anionic polymerization, block copolymers based on olefin monomers containing carbazole groups were also obtained [30]. In this paper, the synthesis and characterization of di-block copolymers polypropylene oxide (PPO) and PEPK are reported. PPO-*b*-PEPK copolymer samples (see Figure 1) were synthesized in one pot via sequential monomer (propylene oxide (PO) and 9-(2,3-epoxypropyl) carbazole (EPK)) addition by using *i-*PrONa and *i-*Bu_3_Al as initiating system. It is worth noting that PO anionic polymerization initiated by alkali metal alkoxides mainly produces oligomeric species, some of which exhibit terminal allyl unsaturations [31,32,33].

Due to the high basicity of propagating species, a proton abstraction from the PO methyl group may occur, resulting in a transfer reaction to monomers, as shown in Scheme 1. However, this side reaction, leading to PO oligomeric species, can be avoided by using initiating systems formed by combining an alkali metal alkoxide with trialkylaluminum [34]. The latter, acting as an activator of PO, not only through the oxygen coordination onto the electrophilic aluminum center, but also increasing selectivity toward active species, would favor the ring opening polymerization reaction rather than the monomer transfer reaction (see Scheme 1) [34].

As a matter of fact, Deffieux et al. reported a controlled anionic polymerization of PO by using *i-*PrONa/i-Bu_3_Al as an initiating system, which provided 100% conversion in a very short time [34]. The observed experimental PPO molar masses were very close to the theoretical values calculated assuming the formation of one polymer chain per *i-*PrONa and dispersity values were also quite narrow [34]. Therefore, starting from the idea that anionic initiators based on alkali metal alkoxides and trialkylaluminum can lead, at least in suitable experimental conditions, to the polymerization of PO presenting living character, we applied this polymerization strategy to prepare PPO-*b*-PEPK copolymers with the aim to obtain a precise control of each block length. It is worth underlining that PPO-*b*-PEPK copolymers are truly intriguing materials because they combine the properties of a low-toxic and soft material such as PPO with those of PEPK, which shows remarkable photoconductive properties. Furthermore, the PPO-*b*-PEPK copolymers could be also able to form ordered structures through self-assembly processes. The microstructure, physicochemical and optical properties of the obtained di-block copolymers are also reported in the paper.

## 2. Materials and Methods

### 2.1. Materials

All manipulations involving air and moisture sensitive compounds were carried out under an atmosphere of dried and purified nitrogen with standard vacuum-line, Schlenk, or glovebox techniques. Glassware and vials used in the polymerization were dried in an oven at 120 °C overnight and exposed to a vacuum-nitrogen cycle three times. All reagents and solvents were purchased from Sigma-Aldrich s.r.l. (Milan, Italy). Toluene was refluxed over sodium for 48 h and distilled before use, propylene oxide was purified on CaH_2_ and distilled before use, and other reagents were used without further purification. *i*-PrONa was synthesized by reaction of *i*-PrOH with sodium dispersed in toluene. The mixture was reacted at 50 °C for one night and stored over a small excess of sodium.

### 2.2. Synthesis of 9-(2,3-epoxypropyl) Carbazole (EPK)

9-(2,3-Epoxypropyl) carbazole was prepared following a procedure already reported in the literature [35]. In a 500 mL round bottom flask, to a solution of KOH (75.0 mmol, 4.21 g) in 200 mL of N,N-dimethylformamide, 5.02 g (30 mmol) of carbazole were added. Mixture was allowed to stir for 30 min, then the temperature was lowered at 4 °C and epichlorohydrin (60.0 mmol, 4.70 mL) was added dropwise. Temperature was left to raise at 20 °C and the reaction mixture was stirred overnight. Afterwards, water (150 mL) was added to the mixture and a white solid precipitant formed. The raw product was filtered, washed with water (3 x 15 mL) and, finally, crystallized in a mixture of ethyl acetate and hexane (1/2). Yields: 65 %. M.p.: 107.3 °C. ^1^H NMR (400 MHz, CDCl_3_) δ: 2.17 (1 H, t, CH), 1.93 (1H, m, CH_2_O), 2.71 (1H, m, CH_2_O), 3.67 (1H, d, CH_2_N), 4.04 (1H, d, CH_2_N), 7.47-6.61 (8H, aromatic CH) ppm. ^13^C NMR (100.62 MHz, CDCl_3_) δ:138.84, 124.30, 121.01, 118.64, 117.0, 107.15, 49.00, 43.58, 42.87 ppm.

### 2.3. PPO and PPO-b-PEPK Synthesis

The polymerization procedure reported below refers to samples **1** and **2** of Table 1. In a 50 mL glass reactor equipped with a magnetic stirring bar, 3.0 mL of PO (42.9 mmol) was dissolved in 6.0 mL of toluene under nitrogen atmosphere. The reactor was thermostated at 0 °C and 0.25 mL of *i*-PrONa solution in toluene (0.45 M, 0.11 mmol) was added to the reaction mixture. After fifteen minutes, 0.86 mL of tri-isobutyl aluminum (1 M in toluene, 0.86 mmol) was added to start the polymerization. The system was maintained under stirring in the thermostated bath. After 3.5 h, half of the reaction mixture was transferred to another glass reactor and 0.5 mL of ethanol was added to stop polymerization. Solvent was then evaporated under vacuum, and the polymer was dried until it reached a constant weight. (Yield: 99%) As for the remaining half of the reaction mixture, the system was heated up to room temperature (20 °C) and 0.478 g of EPK (2.1 mmol) was added. The system was maintained under stirring for 3.5 h and then 0.5 mL of ethanol was added to stop polymerization. The solvent was then evaporated under vacuum, and the polymer was dried until constant weight. The recovered polymer sample was finally washed 3 times with cold pentane. (Yield: 99%)

### 2.4. Methods

*NMR:* NMR spectra of the monomer were recorded on Bruker Advance 400 spectrometer (^1^H, 400 MHz; ^13^C, 100.62 MHz) operating at 298 K. The sample was prepared by dissolving 5 mg of monomer in 0.5 mL of deuterated chloroform (CDCl_3_). Tetramethylsilane (TMS) was used as internal chemical shift reference. NMR spectra of polymers were recorded on a Bruker Advance 600 spectrometer (^1^H, 600.13 MHz; ^13^C, 150.92 MHz) operating at 298 K. The samples were prepared by dissolving 10 mg of polymer in 0.5 mL of CDCl_3_. Tetramethylsilane (TMS) was used as internal chemical shift reference.

Gel permeation chromatography (*GPC*)*:* The molecular weights (M_n_ and M_w_) and dispersity (Ð) of polymer samples were measured by gel permeation chromatography (GPC) at 30 °C, using tetrahydrofuran (THF) as solvent, an eluent flow rate of 1 mL/min, and narrow polystyrene standards as reference. The measurements were performed on a Waters 1525 binary system equipped with a Waters 2414 RI detector using four Styragel columns (range 1000–1,000,000 Å).

*DSC:* Calorimetric measurements were carried out on a DSC Q20 apparatus manufactured by TA Instruments in flowing N_2_. Polymer samples of 5–10 mg were placed in aluminum pans and heated/cooled at a rate of 10 °C/min. Measurements were taken in the range −80 to 200 °C.

UV–vis and fluorescence analysis: UV–vis measurements were performed by a Varian Cary 50 spectrophotometer and photoluminescence was recorded by a Varian Cary Eclipse spectrophotometer. Thin polymer films were prepared by spin coating on a quartz slide substrate. The film thickness and roughness were measured by a KLA Tencor P-10 surface profiler. Fluorescence measurements in solutions were performed in THF.

## 3. Results and Discussion

PPO-*b*-PEPK block copolymer samples were obtained by highly selective one-pot reaction adding sequentially PO and EPK monomers to the polymerization system. For the sake of clarity, the polymerization reaction is depicted in Scheme 2. Experimental polymerization conditions and copolymer features are shown in Table 1. Sequential polymerizations were carried out with different EPK/*i*-PrONa ratios, as reported in Table 1, and the molecular weights of copolymers, both theoretical and experimental, increased with an increasing EPK/*i*-PrONa ratio, indicating a living polymerization.

PPO-*b*-PEPK copolymer microstructures were evaluated by ^1^H and ^13^C NMR analyses. As an example, the ^1^H and ^13^C NMR spectra of sample **2** of Table 1 are shown in Figure 2 and Figure 3, respectively. In particular, the characteristic signals of both PO and EPK regioregular sequence units were recognized and assigned in both spectra. Unfortunately, in both spectra, diagnostic signals, proving the effective formation of PPO-*b*-PEPK copolymer rather than the presence of PPO and PEPK homopolymers, are not detectable. In order to demonstrate the successful sequential polymerization of PO and EPK monomers, a DOSY NMR experiment was set up. In detail, a DOSY NMR experiment of sample **2** was achieved in dilute CDCl_3_. The DOSY map with the related ^1^H NMR spectrum projected on top is reported in Figure 4. The ^1^H NMR spectrum exhibits signals related to both PPO (δ = 3.56, 3.41, 1.15 ppm) and PEPK (δ = 7.89 ‒ 6.89 (aromatic), 3.05, 2.52, 2.42 ppm) blocks. As shown on the DOSY map, ^1^H NMR signals of PPO and PEPK present the same diffusion coefficient—3.99 × 10^−11^ m^2^ s^−1^, which is consistent with an efficient sequential polymerization. Moreover, to prove that PPO growing chains totally reacted with EPK monomer and no residual PPO homopolymer was present in sample **2**, a dilute CDCl_3_ solution of sample **1** (PPO homopolymer) of Table 1 was analyzed by DOSY NMR and compared to sample **2** (PPO-*b*-PEPK block copolymer) spectrum. The diffusion coefficient of sample **1** was found to be 1.70 × 10^−9^ m^2^ s^−1^, different from the one of sample **2,** thus proving that no unreacted PPO homopolymer is present and that the formation of the copolymer was successful.

In addition to the NMR evidence, PPO and PPO-*b*-PEPK samples showed monomodal curves in the GPC analysis (see Figure 5). Molecular weights are quite close to theoretical values, calculated assuming the formation of one polymer chain *per i-*PrONa (see Table 1). Moreover, GPC curves also show very narrow dispersity values. Consistent with those already reported in [34], polymerization seems to proceed with living character without any significant contribution of the monomer transfer process.

From thermal analysis, all PPO-*b*-PEPK copolymer samples showed decomposition temperatures of 330 °C (5% weight loss), while DSC measurements showed two distinct *T*_g_ values, corresponding to the PPO and PEPK blocks, respectively. It is worth underlining that the two *T*_g_ values observed for each single copolymer seem to depend on the length of each block, as reported in Table 1. In fact, while the *T*_g_ of the PPO block remains unchanged in all copolymer samples, consistent with the similar PPO block length in each copolymer, the *T*_g_ values relative to the PEPK block increase with increasing block lengths. As an example, in Figure 6, TGA and DSC curves of PPO-*b*-PEPK sample **2** are reported.

The optical properties of the obtained PPO-*b*-PEPK copolymer samples were also investigated. As an example, the UV–vis spectrum of PPO-*b*-PEPK copolymer sample **2** is shown in Figure 7. The peaks at 330 and 344 nm are distinctive of carbazole and are always observed in the absorption spectra of carbazole-containing polymers [36,37].

The fluorescence spectrum of a THF solution of PPO-*b*-PEPK copolymer (sample **2**, Table 1), measured under an excitation wavelength of 290 nm at room temperature, is reported in Figure 8A. Only two bands in the high energy region presenting maxima at 355 and 367 nm were detected. Differently, the fluorescence spectrum of PPO-*b*-PEPK copolymer sample **2** film, in the solid state, presents a broad band at 400 nm with two shoulders at 354 and 375 nm (see Figure 8B). As expected, both spectra are very similar to those expected for the PEPK homopolymer. It is worth recalling that the optical properties of EPK homopolymers and oligomers [38,39], as well as other polymers containing carbazole pendant groups, have been already deeply investigated [2,3,4,5,20]. It is well known the peculiarity of the photophysical behavior of this class of polymers is able, in some cases, to show excimer fluorescence [40,41,42,43]. In particular, it was reported that polymers with pendant carbazolyl groups do not exhibit excimer emissions in diluted solutions at room temperature, except when the carbazole chromophores are directly bound to the polymer backbone as in the case of PVK [1,40,41,42,43]. On the contrary, in the solid state, carbazole-containing polymers also present a short spacer between the polymer chain and the carbazole groups give rise to excimer fluorescence [1,2,15,18,19,20,21,40,41,42,43]. Moreover, it was reported that, depending on polymer chain conformation, two different excimer fluorescences can be generated: the low energy ‘‘sandwich like” excimer and/or the high energy ‘‘partially overlapping” excimer fluorescence (see Scheme 3) [2,15,18,19,20,21,40,41,42,43].

Consistent with the literature concerning the PEPK homopolymer photoluminescent feature [1], the fluorescence spectra of diluted solutions of PPO-*b*-PEPK copolymers do not exhibit excimer emission (see Figure 8A): the observed bands at 354 and 375 nm, have to be associated to the isolated carbazole fluorescence of PEPK block. Conversely, PPO-*b*-PEPK copolymers in the solid state as thin films seem to show excimer emissions. In our opinion, the broad band observed at 400 nm should be associated to excimer fluorescence, probably, to the low energy ‘‘sandwich like” excimer fluorescence. As already reported for other polymers containing pendant carbazole groups [15,18,19,20,21], these experimental results can be explained assuming that polymer chain conformational freedom in solution prevents excimer formation, while in the solid state, a reduced conformational freedom allows the excimer formation.

## 4. Conclusions

PPO-*b*-PEPK copolymers have been synthesized by using *i*-PrONa/*i*-Bu_3_Al as an initiating system in one pot via sequential monomer addition. This synthetic approach allows the length of each block to be easily controlled by modulating the ratio initiator/monomer in the reaction mixture. The formation of copolymer structures rather than homopolymer mixtures, when starting from a PPO block, was proved by NMR investigations on the diffusion coefficient of the PPO homopolymer compared to the one of PPO-*b*-PEPK block copolymer. Interestingly, both the copolymer molecular weights, quite close to theoretical values calculated, indicate the formation of one polymer chain per *i-*PrONa, and the narrow dispersity values seem to indicate that polymerization proceeds with living character. Altogether, the findings here open the route for an efficient, economic synthesis of copolymers whose blocks can be easily modulated depending on the desired final material. With respect to PPO-*b*-PEPK copolymers, they show intriguing “calorimetric” properties since they consist of two blocks based on homopolymers with very different *T*_g_ values. Such *T*_g_s values are both reproduced in the final copolymeric products making the latter particularly interesting as compatibilizers. As for optical properties, copolymer solutions show only isolated carbazole fluorescence, whereas in the solid state, PPO-*b*-PEPK film samples give rise to excimer emission.

## Data Availability

The data presented in this study are available on request from the corresponding author.

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
