# Peer review of "Synthesis of Di-Block Copolymers Poly (Propylene oxide)-block-Poly (9-(2,3-epoxypropyl) Carbazole) via Sequential Addition of Monomers in One Pot"

_polymers, 2021, doi:10.3390/polym13050763_

Round 1

Reviewer 1 Report

The current manuscript present the synthesis of di-block copolymers polypropylene oxide-b-poly(9-(2,3-epoxypropyl)carbazole) via sequential addition of monomers in one pot. The manuscript is well written, only moderate english changes are required, for instance as for the sentence below, where two verbs are present (indicated in italic).

To thermal analysis, all PPO-b-PEPK copolymer samples showed a decomposition 206 temperature of 330 °C (5% weight loss), while DSC measurements show two distinct Tg, 207 corresponding to the PPO and PEPK blocks respectively, have been detected.

Upon  editing of English language, i will recommend to publish the manuscript.

Author Response

The authors wish to thank editor and all reviewers for their useful comments which contributed to improve the quality of the paper.

The author answers to the reviewer comments are reported below:

Reviewer 1 comments:

The current manuscript present the synthesis of di-block copolymers polypropylene oxide-b-poly(9-(2,3-epoxypropyl)carbazole) via sequential addition of monomers in one pot. The manuscript is well written, only moderate english changes are required, for instance as for the sentence below, where two verbs are present (indicated in italic).

To thermal analysis, all PPO-b-PEPK copolymer samples showed a decomposition 206 temperature of 330 °C (5% weight loss), while DSC measurements show two distinct Tg, 207 corresponding to the PPO and PEPK blocks respectively, have been detected.

Author answer:

We apologize for the mistake. The verb: “have been detected” was deleted from the sentence: “To thermal analysis…”. Moreover, we have carefully revised the manuscript to ensure that the text is optimally phrased and free from typographical and grammatical errors.

Reviewer 2 Report

Authors present a manuscript on the preparation of PPO-b-PEPK diblock copolymer in one pot, by anionic polymerization. Synthesis procedure is well explained in a way that can be reproduced. Classical characterization techniques employed in characterization of synthesized copolymers, such as NMR, GPC, TGA and DSC or UV-vis spectroscopy, have been correctly used an presented. Obtained conclusions seem to be in agreement with obtained data. However, there are some aspects to be considered before accepting the paper:

Language must be improved through the text. Especially in the tittle and abstract there are some errors:

-Tittle: instead of ...diblock copolymers..... in my opinion is better :polypropylene ox-2 ide-b-poly(9-(2,3-epoxypropyl)carbazole) diblock copolymers

-line 14 abstract: instead of used ratio monomer/catalyst it should be monomer/catalyst ratio used. In the same way, copolymer molecular weights should be corrected by molecular weight of copolymers...

-line 17: As for copolymers, their calorimetric properties are remarkably, since they maintain the very different glass transition temperatures typical of the two corresponding homopolymers. This sentence is confusing and must be changed

line 19: to optical analysis... must be changed

Those are only some examples in the tittle and abstract, but severe english correction should be done for all the text.

Moreover, all characterization graphs shown correspond to sample 2 in table 1. Why the rest are not shown? As examples, of course not all of them should be presented, but it is possible to use some of them as example in different characterizations....it is only a suggestion

Author Response

The authors wish to thank editor and all reviewers for their useful comments which contributed to improve the quality of the paper.

The author answers to the reviewer comments are reported below:

Reviewer 2 comments:

General comments

Authors present a manuscript on the preparation of PPO-b-PEPK diblock copolymer in one pot, by anionic polymerization. Synthesis procedure is well explained in a way that can be reproduced. Classical characterization techniques employed in characterization of synthesized copolymers, such as NMR, GPC, TGA and DSC or UV-vis spectroscopy, have been correctly used an presented. Obtained conclusions seem to be in agreement with obtained data. However, there are some aspects to be considered before accepting the paper:

Language must be improved through the text. Especially in the tittle and abstract there are some errors:

  1. Reviewer 2 comment: Tittle: instead of ...diblock copolymers..... in my opinion is better :polypropylene ox-2 ide-b-poly(9-(2,3-epoxypropyl)carbazole) diblock copolymers

Author answer: "Polypropylene oxide" was replaced with "Polypropylene ox-2-ide" in the title of the manuscript.

  1. Reviewer 2 comment: line 14 abstract: instead of used ratio monomer/catalyst it should be monomer/catalyst ratio used. In the same way, copolymer molecular weights should be corrected by molecular weight of copolymers...

Author answer: "used ratio monomer/catalyst" was replaced with: " monomer/catalyst ratio used". (pag 1, line14).

  1. Reviewer 2 comment: line 17: As for copolymers, their calorimetric properties are remarkably, since they maintain the very different glass transition temperatures typical of the two corresponding homopolymers. This sentence is confusing and must be changed

Author answer: " The sentence has been modified as reported: “The calorimetric properties of copolymers obtained in this work are remarkably, since they show two distinct Tg, corresponding to the PPO and PEPK blocks.” (Pag 1, lines 17-19) 

  1. Reviewer 2 comment: line 19: to optical analysis... must be changed

Author answer: It is now reported: “The optical measurements of di-block copolymers show analogous features than those of PEPK homopolymer.” (Pag 1, lines 19-20)

  1. Reviewer 2 comment: Those are only some examples in the tittle and abstract, but severe english correction should be done for all the text.

Author answer: we have carefully revised the manuscript to ensure that the text is optimally phrased and free from typographical and grammatical errors.

  1. Reviewer 2 comment: Moreover, all characterization graphs shown correspond to sample 2 in table 1. Why the rest are not shown? As examples, of course not all of them should be presented, but it is possible to use some of them as example in different characterizations....it is only a suggestion

Author answer: We chose to show 1H, 13C  and 2D Dosy NMR spectra (Figure 2, 3 and 4), TGA and DSC curves (Figure 6) and absorption/emission spectra (Figure 7 and 8) of just one sample as an example, because, the analysis of all other samples are really similar and do not enrich the paper. In author opinion, all figures reported in the manuscript relative to sample 2 are exhaustive to show properties and feature of all PPO-b-PEPK polymers obtained in this work.

Reviewer 3 Report

The paper entitled “Synthesis of di-block copolymers polypropylene oxide-b-poly(9-(2,3-epoxypropyl)carbazole) via sequential addition of monomers in one pot” by Izzo et al. deals with the synthesis and characterization of a series of di-block copolymers based on PPO and PEPK.

The article is clear, well written and the conclusions are supported by the results. However, some corrections are needed in order to increase the quality of the manuscript.

  1. The originality of the study it’s not highlighted in the introduction section.
  2. Line 103: please modify the title “PPO and PPO-b-PEPK synthesis”. Also in this paragraph please add some information about the synthesis of sample 3 and 4.
  3. Line 119: replace “measurements” with “methods”
  4. Line 145-146: the term copolymerization is misused. The authors have carried out a sequential polymerization (addition of the second monomer after the polymerization of first one) and not a copolymerization when the monomers are polymerized simultaneously. Please verify within the all manuscript.
  5. The DOSY spectrum for sample 1 must be provided in supporting information
  6. The results concerning the thermal analysis are very poorly discussed. Both DSC and TGA curves for all the samples must be provided. The authors must indicate the correlation between the Tg values and sequence nature. The increase of the Mn(cop) has no influence on these values?!
  7. In fig 7 the results for PEPK homopolymer should be added
  8. Line 27: “properties that they exhibit”
  9. Line 58: “formed by combining”
  10. Line 76: replace “chemical-physical” with “physicochemical”
  11. Caption of Scheme 2: PPO-b-PEPK copolymer synthesis
  12. Increase the resolution of fig 4
  13. Line 208: delete “have been detected”

In view of the above I recommend the publication of this manuscript in Polymers after major corrections.

Author Response

The authors wish to thank editor and all reviewers for their useful comments which contributed to improve the quality of the paper.

The author answers to the reviewer comments are reported below:

Reviewer 3 comments:

General comments The paper entitled “Synthesis of di-block copolymers polypropylene oxide-b-poly(9-(2,3-epoxypropyl)carbazole) via sequential addition of monomers in one pot” by Izzo et al. deals with the synthesis and characterization of a series of di-block copolymers based on PPO and PEPK.

The article is clear, well written and the conclusions are supported by the results. However, some corrections are needed in order to increase the quality of the manuscript.

  1. Reviewer 3 comment: The originality of the study it’s not highlighted in the introduction section.

Author answer: We thank the referee for the useful suggestion. We modified the introduction part of the paper in an attempt to make it more incisive and clear. In detail, the sentences: “It is worth underlining that PPO-b-PEPK copolymers are truly intriguing materials because they combine the properties of a low-toxic and soft material such as PPO, with those of PEPK which shows remarkable photoconductive properties. Furthermore, the PPO-b-PEPK copolymers could be also able to form ordered structures through self-assembly processes.” were added in the text (pp.2,3; lines 76-80).

  1. Reviewer 3 comment: Line 103: please modify the title “PPO and PPO-b-PEPK synthesis”. Also in this paragraph please add some information about the synthesis of sample 3 and 4.

Author answer: Title was modified.

The copolymerization procedure for all the other samples is completely similar to that reported for sample 2. The copolymerization tests differ only in the amount of EPK used. These data are reported in Table 1 of manuscript.

  1. Reviewer 3 comment: Line 119: replace “measurements” with “methods”

Author answer: the term: “measurements” was replaced with “methods” (pag 3, line 123) 

  1. Reviewer 3 comment: Line 145-146: the term copolymerization is misused. The authors have carried out a sequential polymerization (addition of the second monomer after the polymerization of first one) and not a copolymerization when the monomers are polymerized simultaneously. Please verify within the all manuscript.

Author answer: we agree with reviewer 3 and apologize for the mistake. We revised the manuscript and the term: "copolymerization" was replaced with "sequential polymerization".

  1. Reviewer 3 comment: The DOSY spectrum for sample 1 must be provided in supporting information

Author answer: We thank the reviewer for the suggestion, however we decided to maintain figure 4 in the manuscript, because this analysis, 2D DOSY NMR, proves that the polymerization product is indeed a block copolymer.

  1. Reviewer 3 comment: The results concerning the thermal analysis are very poorly discussed. Both DSC and TGA curves for all the samples must be provided. The authors must indicate the correlation between the Tg values and sequence nature. The increase of the Mn(cop) has no influence on these values?!

Author answer: We agree with the reviewer. The Tg values of all samples were added in table 1. Moreover, in the manuscript a short discussion on this was added. In detail the sentences: " It is worth underlining that the two Tg values observed for each single copolymer seem to depend on the length of each block as reported in Table 1. In fact, while the Tg of the PPO block remains unchanged in all copolymer samples, consistently with the similar PPO block length in each copolymer, the Tg values relative to PEPK block increase with increasing of the block lengths. As an example, in Figure 6, TGA and DSC curves of PPO-b-PEPK sample 2 are reported." were added in the text (pag. 7, lines 212-217)

  1. Reviewer 3 comment: In fig 7 the results for PEPK homopolymer should be added

Author answer: We thank the reviewer for the suggestion, however the UV-vis spectrum of PEPK does not add any important information to the work and it has already been widely reported in the literature (see i.e. Polym Int 57:1159–1164 (2008)). For reviewer requirement, we report here the UV-vis spectrum of a PEPK sample in comparison with that of PPO-b-PEPK copolymer.

UV-vis spectra of PPO-b-PEPK and PEPK

  1. Reviewer 3 comment: Line 27: “properties that they exhibit”

Author answer: The sentence has been modified as suggested

  1. Reviewer 3 comment: Line 58: “formed by combining”

Author answer: “by” has been added

  1. Reviewer 3 comment: Line 76: replace “chemical-physical” with “physicochemical”

Author answer: The term has been changes as suggested

  1. Reviewer 3 comment: Caption of Scheme 2: PPO-b-PEPK copolymer synthesis

Author answer: Caption has been changes as suggested

  1. Reviewer 3 comment: Increase the resolution of fig 4

Author answer: we improved quality of Figure 4

  1. Reviewer 3 comment: Line 208: delete “have been detected”

 Author answer: The sentence has been corrected

Round 2

Reviewer 3 Report

The article can be publish as it is.

Author Response

All authors thank the reviewer for his helpful suggestions 

This manuscript is a resubmission of an earlier submission. The following is a list of the peer review reports and author responses from that submission.

Round 1

Reviewer 1 Report

This draft presented a synthesis of polypropylene oxide-b-2 poly(9-(2,3-epoxypropyl) carbazole) diblock copolymers. However, I find this draft needs a significant amount of work to be accepted in this journal. I suggest the authors very very carefully revise this draft before submission. I, therefore, suggest reject. My comments are listed as follows.

  1. The introduction part is very confusing. What is the rationale to synthesize such a diblock system? Where is the novelty of this draft comparing with published work?
  2. line 47-56, this part of content seems to be a part of "results and discussion" section instead of "introduction".
  3. There isn't even a title for scheme 1. This is unacceptable!
  4. Table 1, you showed 4 runs of copolymerization. Please present the GPC curves of these runs.
  5. line 154, I don't understand  "diagnostic signals of the junction 154 between the PPO and PEPK blocks were unfortunately". What is this "junction"?
  6. please show 1H-NMR. In a typical characterization of diblock copolymers, 1H-NMR is usually showed instead of 13C-nmr.
  7. Figures in this draft are fuzzy and unacceptable for publications.
  8. Figure 4, explain the low molecular weight tailing of the GPC traces.

Reviewer 2 Report

The authors describe anionic polymerization of glycidyl monomers. 

Scheme 1. text above arrow: 'transfer' can be revised to 'chain-transfer'.

Line 115: remove underline from the text 'deuterated chloroform'.

Figure 2: Please make it full width and remove the table with chemical shift numbers. Please increase signal intensity.

There seems to be a shoulder in the GPC data (Figure 4). Please explain.

A number of text lines in the references are underlined.

If ethanol is added for quenching polymerization, why not show the full chemical structure of the final polymer in scheme 2 (add the quenching step and write the neutral polymer structure).

Why not show a few example of end-functional polymers by adding different polymerization quenchers. Such an addition will increase the quality and impact of the work.

Reviewer 3 Report

polymers-1084463

The article: “Synthesis of di-block copolymers polypropylene oxide-b-poly(9-(2,3-epoxypropyl) carbazole) via sequential addition of monomers in one pot” by L. Izzo et al. describes the one-pot synthesis of polypropylene oxide (PPO) and poly(9-(2,3-epoxypropyl) carbazole) (PEPK) di-block co-polymers via sequential monomer addition utilizing i-PrONa/i-Bu3Al as catalytic system. Moreover, a series of characterization techniques were carried out to prove the successful formation of the diblock copolymers and the study of optical and emission properties.

Generally, the MS is well-written and provides to the readers a detailed experimental description of the procedure followed for the synthesis of the final diblock copolymers. The length of the manuscript is relatively small; the authors could include more details in the results and discussion section, especially regarding the DSC measurements. I suggest that this work could be published in Polymers. Some comments were raised by reading the manuscript:

  • Why the authors choose PPO as the first block of the final copolymer? Were they interested in some specific properties in the final copolymer with PEPK?
  • Regarding the polymerization of the second block (PEPK), how the authors know and stopped the polymerization after 3.5h? Did they conduct any kinetic studies of PEPK with this catalytic system?
  • The authors mentioned that molecular weights are relatively close to theoretical values (Table 1). They performed GPC measurements where PS standards were used for calibration, so the Mn values of the polymers are not the exact ones. They are apparent Mn. Also, I suggest to the authors to include in Table 1 the apparent Mn of the first block (PPO), along with the final Mn diblock copolymers.
  • Figure 4: In the GPC trace of PPO block, a small shoulder appears in the region of low molecular weights. Can the authors comment on this? Similar products appear in the final diblock copolymer.
  • Line 190-192: The authors should include in the MS the DSC thermograph of the corresponding copolymer and comment more about the thermal properties of this system.

Also, a few minor comments:

  • Line 136-140 were also described in the experimental section, no need to repeat again the procedure in the results and discussion section
  • No caption in Scheme 1, 2, 3.